# Impact of processing method on donated human breast milk microRNA content

**Urszula Smyczynska**[ID][1☯], **Marcin A. Bartlomiejczyk**[2☯], **Marcin M. Stanczak**[1],
**Pawel Sztromwasser**[ID][1], **Aleksandra Wesolowska**[3], **Olga Barbarska**[3],
**Emilia Pawlikowska**[4], **Wojciech Fendler**[ID][1,5]*

**1** Department of Biostatistics and Translational Medicine, Medical University of Lodz, Lodz, Poland,
**2** Department of Hypertension, Medical University of Lodz, Lodz, Poland, **3** Laboratory of Human Milk and
Lactation Research, Department of Neonatology, Medical University of Warsaw, Regional Human Milk
Bank in Holy Family Hospital, Warsaw, Poland, **4** Institute of High Pressure Physics, Polish Academy of
Sciences, Warsaw, Poland, **5** Department of Radiation Oncology, Dana-Farber Cancer Institute, Boston,
Massachusetts, United States of America

☯ These authors contributed equally to this work.
* wojciech.fendler@umed.lodz.pl

org/10.1371/journal.pone.0236126

CHINA

**Data Availability Statement:** Data are deposited in
Gene Expression Omnibus (GEO) under accession
number GSE142282 (https://www.ncbi.nlm.nih.
gov/geo/query/acc.cgi?acc=GSE142282).

## Abstract

Pasteurization of donated human milk preserves it for storage and makes it safe for feeding,
but at the expense of its composition, nutritional values and functions. Here, we aimed to
investigate the impact of Holder Pasteurization (HoP) and High Pressure Processing (HPP)
methods on miRNA in human milk and to evaluate impact of these changes on miRNA func-
tions. Milk samples obtained from women in 50th day of lactation (n = 3) were subjected
either to HoP, HPP or remained unpasteurized as a control. Subsequently, miRNA was iso-
lated from whole material and exosomal fraction and sequenced with Illumina NextSeq 500.
Sequencing data were processed, read counts were mapped to miRNA and analyzed both
quantitatively with DESeq2 and functionally with DIANA mirPath v.3. While HPP caused sta-
tistically insignificant decrease in number of miRNA reads compared to unprocessed mate-
rial, HoP led to 82-fold decrease in whole material (p = 0.0288) and 302-fold decrease in
exosomes (p = 0.0021) not leaving enough reads for further analysis. Changes in composi-
tion of miRNA fraction before and after HPP indicated uneven stability of individual miRNAs
under high pressure conditions, with miR-30d-5p identified as relatively stable and miR-29
family as sensitive to HPP. Interestingly, about 2/3 of unprocessed milk miRNA content con-
sists of only 10 distinct miRNAs with miR-148a-3p at the top. Functional analysis of most
abundant human milk miRNAs showed their involvement in signaling pathways, cell com-
munication, proliferation and metabolism that are obviously important in rapidly growing
infants. Functions of miRNAs which suffered the greatest depletion during HPP were similar
to roles of the majority of unprocessed human milk's miRNA, which indicates that those
functions may be weakened although not completely lost. Our findings indicate that HPP
is less detrimental to human milk miRNAs than HoP and should be considered in further
research on recommended processing procedures for human milk banks.

**Funding:** This research was funded by the National Science Center in Poland (https://ncn.gov.pl/) PRELUDIUM grant number UMO 2016/21/N/NZ2/01726 (MAB) and the National Science Center in Poland POLONEZ grant number 2016/23/P/NZ2/04251 (PS). This project has received funding from the European Union's Horizon 2020 (https://ec.europa.eu/programmes/horizon2020/en) research and innovation programme under the Marie Skłodowska-Curie grant agreement No 665778 (PS). The funders had no role in study design, data collection and analysis, decision to publish, or preparation of the manuscript.

**Competing interests:** The authors have declared that no competing interests exist.

## Introduction

Breast milk is uniquely tailored for the human infant, both in terms of nutritional composition and in the non-nutritive bioactive factors that promote survival and healthy development [1]. It influences development and maturation of infant's organs and tissues in many ways. It shapes the composition of microbiotic flora of the neonate, indirectly influencing the immune system's function. It also directly "educates" the neonatal immune system to react appropriately upon microbial and antigenic challenges [2]. Human milk contains the required nutritional elements for the infant, including carbohydrates, proteins, lipids, and minerals, as well as bioactive factors which boost infants' immunocompetence and serve developmental functions [3].

MicroRNAs (miRNAs)—abundantly present in human milk—have recently been postulated to belong to the later class of substances participating in the regulation of immunological and developmental processes [4, 5]. The role of milk miRNAs is a subject of ongoing research [6] with two competing hypotheses: *functional* stating that they are capable of exerting regulatory impact on infants' organism and *nutritional* claiming that they are simply a source of nutrition. The former hypothesis requires that miRNA are uptaken in the intestines rather than degraded by digestive fluids. Although Title et al. in 2015 found no sign of miRNA uptake from ingested milk in mice [7], Baier et al. [8] provided evidence that even amounts of miRNA absorbed from 0.25 l of cow's milk are sufficient to alter human gene expression and Wang et al. observed changes in bovine miRNA concentrations after ingestion of dietary products [9].

The functional hypothesis can be also supported by numerous studies proving that miRNA content of mammalian milk changes in response to external and internal stimuli. Preterm delivery results in different milk miRNA profile than childbirth on term [10, 11] which is supposed to play a protective role in premature infants [11, 12]. Analysis of porcine milk showed gradual decrease in abundance of immune-related miRNA during lactation period [13]. Changes in mammalian milk miRNA profile can be also induced by infection [14] and maternal diet [15]. Furthermore, Alsaweed et al. [16] identified several human milk cell miRNAs that are endogenously synthesized in the breast and are involved in the synthesis and regulation of milk components such as triacylglycerol, fatty acids, lactose, and others. Thus, with accumulating evidence for their bioactivity, human milk miRNAs are becoming a field of active scientific exploration [5, 17].

A topic of specific interest in this field is the role of exosomes in miRNA delivery and protection [18]. *In vitro* studies of human milk confirmed that exosomes can escape digestion and be absorbed by intestines [10]. Once uptaken, milk exosomes can transfer miRNA to recipient organs, which was proved by Manca et al. in 2018 [19]. In other study, endocytosis of cow milk-derived exosomes by human vascular endothelial cells was suggested as a mechanism of transferring their content to human cells [20].

All the benefits of ingesting breast milk are readily available to healthy newborns fed by their own mothers, but detailed knowledge about breast milk composition and functions is still crucial in management of newborns, particularly in complicated cases, high risk ones and those whom biological mothers cannot breastfeed. According to the Recommendations From the European Milk Bank Association (EMBA) [21], if the mother's own milk is not available for the newborn, donor milk should be given priority over a synthetic substitute. However, in such situation appropriate procedures are introduced in order to make the donation safe and the milk itself preserved for longer period. Nowadays, milk donated to the milk banks is usually pasteurized by the Holder method (62.5°C, 30 min). Unfortunately, it is not an ideal method due to imperfect efficacy in inactivating pathogens in milk and detrimental impact on

proteins and other immunoactive components transferred in the milk [22, 23]. Therefore other, less damaging, methods of pathogen elimination are currently being considered such as: Ultraviolet-C (UV-C) irradiation, High Pressure Processing (HPP) [24] or High-Temperature-Short-Time (HTST) pasteurization [21, 25]. The HPP pasteurization seems to produce the best results in terms of the lowest impact on nutrients found in milk while preserving efficacy in inactivating microorganisms [26, 27]. Other researchers have shown that after processing by HPP, some of the biological activity is still retained, including: lactoferrin, lysozyme, immunoglobulins (A, M and G classes), cytokines (IFN -, EGF, TNF -, TGF -1/ -2) and interleukins (6, 8, 12, 17) or / tocopherol [26, 28–31].

Taking into account all the evidence in favor of miRNA as functional component of breast milk, a search for sterilization methods that preserves their functions seem to be a valid research question. Some experiments indicated that miRNAs in human milk are stable even under harsh conditions, including pH 1, freeze-thaw cycles and treatment with RNase [32] whereas proteins are more vulnerable to heating. However, studies on animal milk showed changes in miRNA profile and abundance depending on milk processing (including both industry and laboratory scale technologies) as well as miRNA isolation method [33, 34]. This has urged us to comprehensively evaluate the total and exosome-bound content of miRNAs in human milk depending on the preservation method used.

## Materials and methods

### Milk samples collection and preparation

Milk samples were obtained from 3 volunteers on the 50th day of lactation (mature milk) in a volume of 150 ml. Each volunteer breastfeed her own healthy child, delivered after full-term single pregnancy (detailed characteristics of milk donors in Table 1). Loss of milk had no effect on newborn feeding. The volunteers were initially recruited as regular milk donors to the Regional Human Milk Bank in Holy Family Hospital in Warsaw, Poland after they gave birth. They fulfilled the conditions required to become a milk donor, including absence of addictions and excluded diabetes (type I, type II and gestational diabetes). Later, they were asked to donate milk samples for this study to which they agreed. They signed an informed consent form to participate in the study, which was approved by the Bioethics Committee of the Medical University of Lodz (consent number is: RNN/01/17/KE dated 17. Jan. 2017). At the time of milk donation they were healthy, not presenting symptoms of any infectious disease.

**Table 1. Characteristics of milk donors, their deliveries and newborns.**

|  | Donor 1 | Donor 2 | Donor 3 |
|---|---|---|---|
| **age** | 35 | 32 | 24 |
| **ethnicity** | white | white | white |
| **pregnancy** | 2nd | 3rd | 2nd |
| **delivery** | 2nd | 1st | 1st |
| **mode of delivery** | vaginal | cesarean | cesarean |
| **Newborn** | | | |
| **sex** | male | male | male |
| **birth weight [g]** | 3460 | 2980 | 4040 |
| **gestational age [weeks]** | 41 | 39 | 40 |
| **Apgar score** | 10 | 10 | 10 |

Immediately after donation, the sample was immediately aliquoted into 3 equal volumes for further processing. Later, they were subjected to the standard HoP and HPP, the third aliquot was unpasteurized milk as a control. Holder pasteurization was performed with automatic Human Milk Pasteurizer S90 Eco (Sterifeed, Medicare Colgate Ltd, England, Cullompton) with the recommended conditions of 62.5˚C for 30 minutes. High pressure processing was performed in 450 MPa for 15 min. Samples were exposed to high pressure treatment at the Institute of High Pressure Physics, Polish Academy of Sciences, using U 4000/65 apparatus (Unipress Equipment, Poland, Celestynow). The maximum pressure available in the apparatus was 600 MPa, the treatment chamber had 0.95 L volume. The pressure-transmitting fluid used was distilled water and polypropylene glycol (1:1). The working temperature of the apparatus ranged from −10˚C to +80˚C. A pressure of up to 600 MPa was generated over 15–25 s; the release time was 1–4 s. Immediately after the processes, the samples were frozen at -20˚C. Summary of milk samples characteristics can be found in S1 Table.

## miRNA isolation, library preparation and sequencing

The purpose of experiment included analysis of milk-derived miRNA isolated using denaturing agents from the whole material and miRNAs specifically extracted from exosomes. Exosomes were isolated from 5 ml of milk using miRCURY Exosome Cell/Urine/CSF Kit (Qiagen, Hilden, Germany). miRNA was isolated with a biofluid-tailored Serum/Plasma Advanced Kit (Qiagen) both from whole material and exosomes. Quality of obtained material was assessed by automatic electrophoresis in TapeStation 2200 (Agilent, USA, Santa Clara) using a HS-RNA kit. Next, the cDNA sequencing libraries were prepared with use of QIAseq miRNA Library Kit (Qiagen), according to manufacturer's protocol. Each sample was marked by a unique molecular index. Library preparation procedure included also the ligation of Unique Molecular Identifiers (UMI) that were later exploited during bioinformatics analysis, particularly quantification of miRNA. Quality of libraries was assessed again using the TapeStation 2200 device (Agilent) with an HS D100 kit. Automatic electrophoresis confirmed the presence of miRNA-sized library (160-175bp) and in some samples additionally a fraction of piRNA-sized library (180-195bp) as usually seen under this protocol. Next, the concentration of RNA in the prepared libraries was measured with a Qubit Fluorometer (Thermo Fisher Scientific, USA, Waltham). Each measurement was repeated twice and the mean concentration of cDNA was presented in S1 Table. Concentration of cDNA in libraries was normalized according to the NextSeq System Denature and Dilute Libraries Guide to 4 nM, with exception of samples Milk-3B and Milk-1B that were normalized to 1 nM and 0.5 nM, respectively, due to initially low concentration of cDNA. Then, all samples were denatured and diluted to a cDNA concentration of 20pM. Single-end sequencing was performed on NextSeq 500 sequencer (Illumina, USA, San Diego) with read length set to 75bp and after final dilution of samples to concentration of 1.8 pM. Experimental data from miRNA sequencing were stored in 18 FASTQ files, three for each of the six processing variants: total and exosomal miRNA isolations and two processing groups (HoP and HPP) and raw milk.

## Sequencing data processing

Read adapters were trimmed and UMI sequences were extracted by UMI-tools v1.0.0 [35]. Maximum 2bp mismatch in 19bp-long adapter sequence was allowed. Reads with incomplete UMI sequence (<12nt) and shorter than 15bp after trimming were excluded from the downstream analyses. Trimmed reads were mapped to miRBase v22 mature human miRNA sequences. Prior to mapping, mature miRNA records with identical sequences were collapsed to allow unique mapping. Bowtie2 v2.3.4.1 [36]. local alignment with following parameters

was used: "-N1 -L9 –norc -k10 –local –score-min L,4,1.3 –mp 4". Reads mapping to multiple miRNAs with equal scores were excluded. Uniquely mapped reads were deduplicated using bwased on UMI using the "unique methods in UMI-tools [35]. Next, SAMtools v1.9 flagstat [37] was used to obtain read counts for each miRNA. Finally, reads not mapped to miRBase, were aligned to human genome reference sequence (hg38) using the Burrows-Wheeler (BWA 0.7.17) alignment algorithm [38] with subsequent feature assignment with featureCounts program from the Subread package v1.6.1 [39]. During data processing, quality control statistics were generated on several levels using FastQC [40], MultiQC [41] and custom scripts.

## Statistical analysis

Read counts and quality control data were subjected to statistical analysis performed with the use of Python statistical libraries. First, we analyzed the differences in read counts and length of inserts between samples subjected to different processing methods in order to detect potential degradation of RNA due to pasteurization. Distributions of read lengths were compared by the $\chi^2$ goodness of fit test. Log-transformed numbers of reads mapped to miRNA were compared by the paired *t*-tests between processed (either HoP or HPP) and unprocessed milk. We assessed also the number and percentage of reads not mapped to the human genome. At this stage we excluded from further analysis all samples with less than 10000 reads mapped to miRBase, subsequently referred to as miRNA reads. In all excluded samples miRNA reads fraction was below 0.2% (0.04-0.17%) of all reads, while for the rest it exceeded 1% in all cases (1.09-25%; S1 Table). Read counts from the qualified samples were then transformed to transcripts per million (TPM) so that samples with different sequencing depths could be compared. Then, we assessed number of distinct miRNAs detectable in at least 10 TPM in both whole material and exosomes before and after processing. Differences in particular miRNA expression between raw and processed milk were analyzed using the DESeq2 tool [42] (implementation available in GenePattern [43]). According to the DESeq2 manual, unnormalized read counts were inputted, together with samples assignment to groups in terms of processing method. We also correlated the composition of exosomal and total miRNAs in raw and HPP samples (as the exosomal fraction was nearly absent in HoP samples). Finally, for both the total miRNA and exosomes we selected most expressed miRNAs which accounted for 90% of total miRNA reads and we submitted them to functional analysis in DIANA mirPath v.3 [44]. We set the tool to use in silico miRNA target prediction algorithm TargetScan [45]. Analogically, we sought functional annotations of a set of miRNAs which were most highly depleted during HPP.

## Results

### Effect of milk-processing method on miRNA content

Total number of short RNA reads ranged between 3,073,433 and 7,733,423 in whole material samples and from 3,367,403 to 30,512,466 in exosomal samples. On average sequencing depth was slightly lower in samples subjected to HoP, but in all cases this was considered sufficient to warrant further analysis (Fig 1A, S1 Table). Filtration of reads with short inserts or incomplete UMI sequences revealed considerable differences between different processing methods. Eligible reads (Fig 1B) in HoP samples constituted less than 10% in 5 samples and about 18% in the 6th one (Milk-2B), while it was on average 47% in the unprocessed samples and 41% in HPP samples. At the same time we observed similarly unfavorable proportion of eligible and rejected reads in all samples of whole milk from donor B (Milk-1B, Milk-2B, Milk-3B), irrespective of processing method. Thus, we decided to exclude those 3 samples from further analysis, considering this to be rather an artifact than true result.

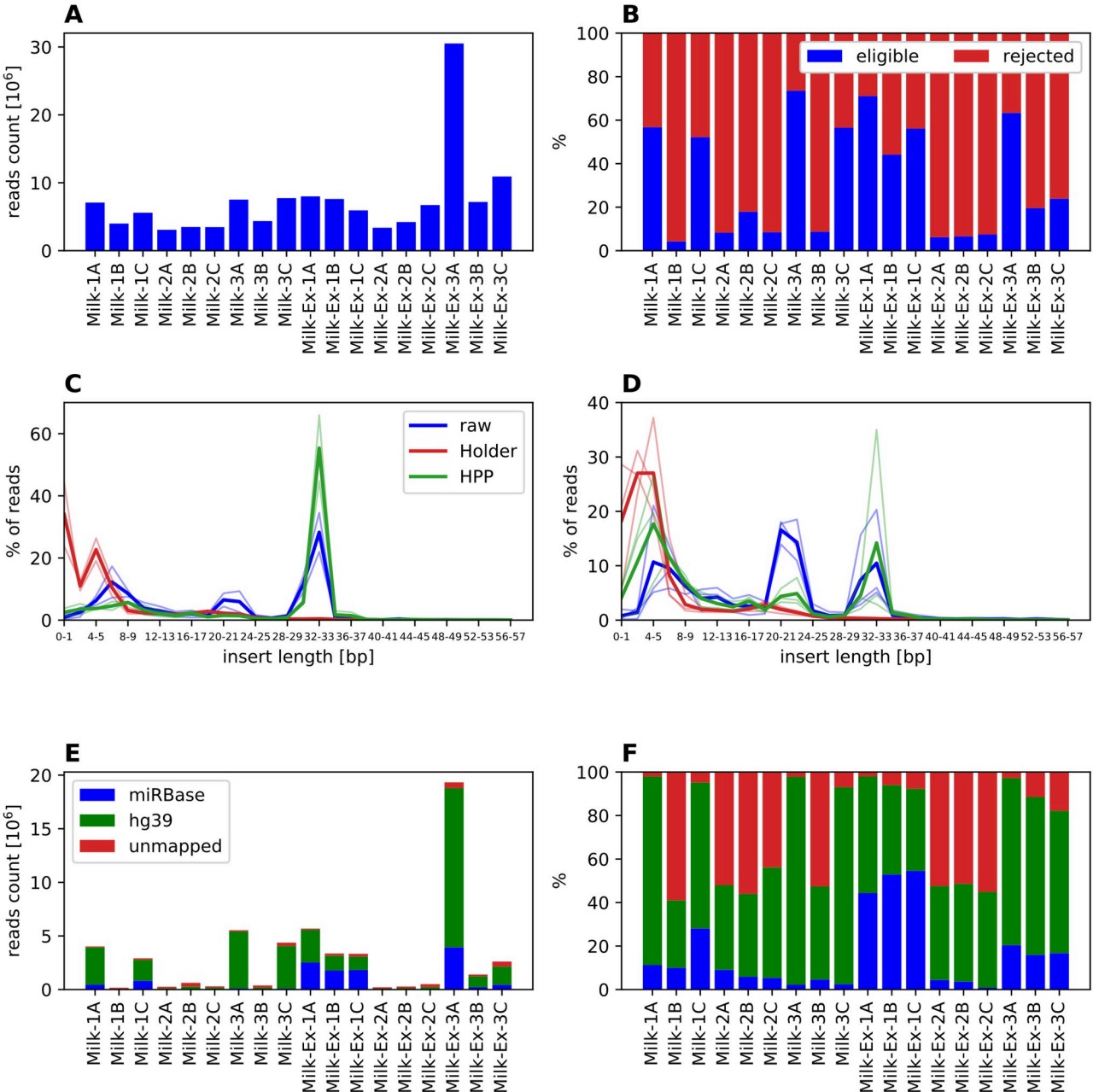

**Fig 1. Reads mapping statistics for milk samples.** A: raw number of reads; B: percentage of reads eligible for mapping (full UMI sequence and insert no shorter than 15bp); C: distribution of insert lengths in whole milk RNA samples, thick line represents mean in group, thin lines single samples; D: distribution of insert lengths in exosomal RNA samples; E: counts of reads mapping to miRBase, human genome (hg38) and not mapped to either; F: percentages of reads mapping to miRbase, human genome (hg38). In panels C-D samples Milk-1B and Milk-3B are excluded.

Next, we investigated the reasons for reads failing the filtering criteria. The dominant problem in HoP samples was insufficient insert length in the reads (Fig 1C and 1D). The distribution of insert lengths differed significantly between unprocessed and HoP samples (p<0.0001 in $\chi^2$ tests for both whole material and exosomes), and similarly between unprocessed and HPP samples (again p<0.0001 for both whole material and exosomes). In RNA isolated from whole, unprocessed milk we observed a dominant peak at 29-36bp—a typical length of

piRNA. The second, lower peak was found at 20-25bp and represented miRNAs. In HPP milk the piRNA-size peak around 32-33bp was preserved, while the miRNA-size peak seemed severely diminished. Reads from HoP-pasteurized milk had mostly very short inserts or no insert at all, indicating high levels of RNA degradation. The distribution of insert lengths from exosome samples is different than in the case of whole material. First, we observe a greater proportion of miRNA size inserts and lower contribution of piRNA size ones. Therefore, it seems that miRNAs are preferentially loaded into extracellular vesicles. Secondly, the proportion of very short inserts is higher than in the case of RNA isolated from whole milk. The effect of processing is similar in whole milk and exosomal fraction; however in the distribution for exosomal insert lengths from HPP milk we can still distinguish a peak corresponding to miRNA size which is not the case for whole material samples.

The number and proportion of reads mapped to miRNA (raw counts given in S2 Table), human genome and unmapped are presented in Fig 1E and 1F, respectively. The low number of reads acceptable as input to mapping tools is visible in all pasteurized samples and in some other samples from donor B (Milk-1B, Milk-3B). Additionally, those samples are characterized by high percentage of unmapped reads, while in all the other samples the majority of reads was mapped either to miRBase or to human genome (hg38). The number of miRNA reads from HoP samples (S1 Table) was 82-fold lower (p = 0.0288) in whole material and 302-fold lower (p = 0.0021) in exosomes than in respective unprocessed milk samples. Similar comparison for HPP milk did not show significant differences (p = 0.2146 for whole material, p = 0.3656 for exosomes), although number of miRNA reads was still 4.2 times higher in whole raw milk and 1.5 times higher in exosomes than in respective HPP samples. The reads not mapped to miRBase, but mapped to the human genome were predominantly fragments of protein-coding RNA, long non-coding RNAs, or fragments without any known biological function (S1 Fig).

## Composition of miRNA fraction

Due to the extremely low number of miRNA reads in milk subjected to thermal processing (less than 10000 reads per sample as shown in S1 Table), the analysis of miRNA fraction composition and function was performed only for raw and HPP milk samples. Additionally all whole material samples from donor B were excluded due to the same reason. Characteristics of samples eligible for further analysis are presented in Table 2.

The lists of miRNAs detectable in different samples were highly overlapping regardless of the processing method or source of miRNA (whole material or exosomes) as presented in Fig 2A. Among miRNAs that passed the established threshold (at least 10 TMP in every replicate)

**Table 2. Number of miRNAs detectable in samples eligible for further analysis.**

| Sample name | Donor | Processing method | Material for RNA extraction | Number of detectable miRNAs | Number of miRNAs with at least 10 reads | Number of miRNAs with at least 10 TPM |
|---|---|---|---|---|---|---|
| Milk-1A | A | None | whole | 538 | 202 | 300 |
| Milk-1C | C | None | whole | 859 | 291 | 355 |
| Milk-3A | A | HPP | whole | 566 | 196 | 377 |
| Milk-3C | C | HPP | whole | 568 | 180 | 568 |
| Milk-Ex-1A | A | None | exosomes | 909 | 333 | 265 |
| Milk-Ex-1B | B | None | exosomes | 983 | 327 | 323 |
| Milk-Ex-1C | C | None | exosomes | 788 | 262 | 312 |
| Milk-Ex-3A | A | HPP | exosomes | 1218 | 419 | 318 |
| Milk-Ex-3B | B | HPP | exosomes | 801 | 238 | 471 |
| Milk-Ex-3C | C | HPP | exosomes | 1016 | 261 | 463 |

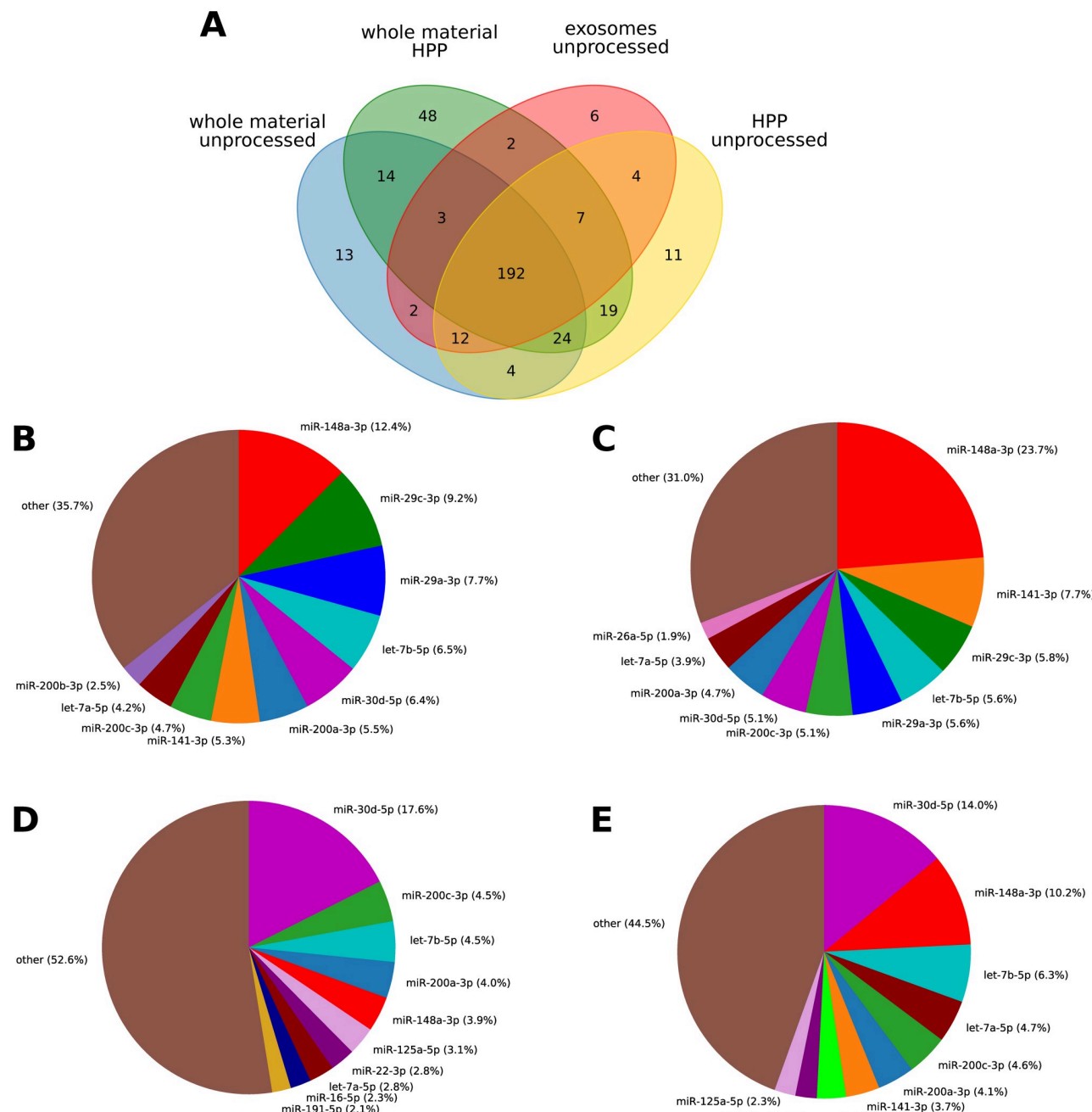

**Fig 2. miRNAs detectable in milk before and after HPP.** A: Venn diagram illustrating number of miRNAs detectable in whole material and exosomes in at least 10 TPM (in every sample in the same group) before and after processing; 10 most abundant miRNAs from whole material (B) and exosomal fraction (C) in unprocessed milk; miRNAs from whole material (D) and exosomal fraction (E) in milk subjected to HPP. Percentages are based on mean TPM from all samples in each group.

192 were common to all analyzed samples. Despite differing in abundance, 9 of the top 10 most abundant raw milk miRNAs overlapped between the whole milk (Fig 2B) and exosomes (Fig 2C). In both cases miR-148a-3p was detected in the greatest amount, accounting for almost 24% of total exosomal miRNA and about 12% in whole milk. HPP caused substantial changes in miRNA fraction composition (Fig 2D and 2E). Percentage of miR-148a-3p dropped

to about 1/3 of its level in raw milk, what may indicate its low stability. After HPP miR-29a-3p and miR-29c-3p dropped out of top 10 in both total and exosomal fraction, while miR-30d-5p replaced miR-148a-3p as the most abundant one in HPP milk. Some top 10 members in unprocessed milk were replaced by other miRNAs after HPP, for instance miR-125a-5p and miR-22-3p emerged both in whole material and exosomal fraction.

## Quantitative analysis of miRNA abundance before and after HPP

Quantitative analysis of changes in miRNA abundance after HPP was performed using DESeq2 (full results in S3 Table). Since the algorithm, dedicated primarily to differential gene expression analysis, includes normalization of read counts under the assumption of an equal number of up- and down-regulated genes (which does not need to be the case for a biofluids of variable concentration) we expectedly observed both decreases and increases in particular miRNAs' quantities (Fig 3A). However, in this case any increases could only be due to the normalization process, possibly bolstered by an increase of raw count numbers for transcripts with initial low abundance in raw milk, when some more abundant miRNA were partially degraded by HPP. For miRNA present in at least 100 TPM, their quantities (log-transformed) in whole material and exosomes correlated nearly perfectly both before (r = 0.9546, p<0.0001) and after HPP (r = 0.9352, p<0.0001), as shown in Fig 3B. Observed correlation (r = 0.7585, p<0.0001) between fold changes in total and exosomal fraction (Fig 3C) indicates that HPP similarly affected all milk miRNAs, regardless of their sequestration in vesicles. The analysis of miRNAs whose quantity changed the most (Fig 3D and 3E) largely confirmed this observation. The greatest statistically significant loss in both types of samples is observed for miR-29c-3p (presented for every donor in Fig 4A), miR-29a-3p (Fig 4B) and miR-378a-3p (Fig 4C).

## Functional analysis of human milk miRNA

Lists of the most abundant miRNAs which accounted for 90% of total reads showed high overlap between whole material and exosomes (Fig 5A, S4 Table). Functional annotations of their targets to KEGG pathways were revealed by analysis performed with DIANA mirPath v.3 (Fig 5B). The most significant pathways were the same for the whole material and exosomes and comprised, among others: ECM-receptor interaction, prion disease, fatty acid biosynthesis and focal adhesion. The most significant pathway, ECM-receptor interaction, contains genes responsible for organ morphogenesis and maintenance of cell and tissue structure. Focal adhesion pathway's constituents play a role in cell proliferation, differentiation and motility. Prion diseases pathway combines genes involved in several pathways leading to neural death, such as activation of compliment and synaptic alternation, which might be involved in immunomodulation and synaptic pruning in children. Involvement of miRNA in fatty acid biosynthesis reflects their impact on selection of source of energy. Pathways distinctly annotated to exosomal miRNA targets included metabolism of xenobiotics, p53 signaling, heparin biosynthesis, fatty acid metabolism and bacterial invasion of epithelial cells. Pathways significantly associated with miRNA content of the whole material miRNA target adherens junction, neurotrophin signaling pathway and signaling regulating pluripotency of stem cells. Next, the impact of HPP on functions of breast milk miRNA was evaluated by identification of functional annotation of 8 miRNAs which suffered the biggest depletion during HPP (Fig 3D). In this case, again, the most significant pathway was ECM-receptor interaction (Fig 5C). Loss was noted also in miRNAs responsible for, among others, regulating protein digestion and absorption, focal adhesion, platelet activation and amoebiasis pathways.

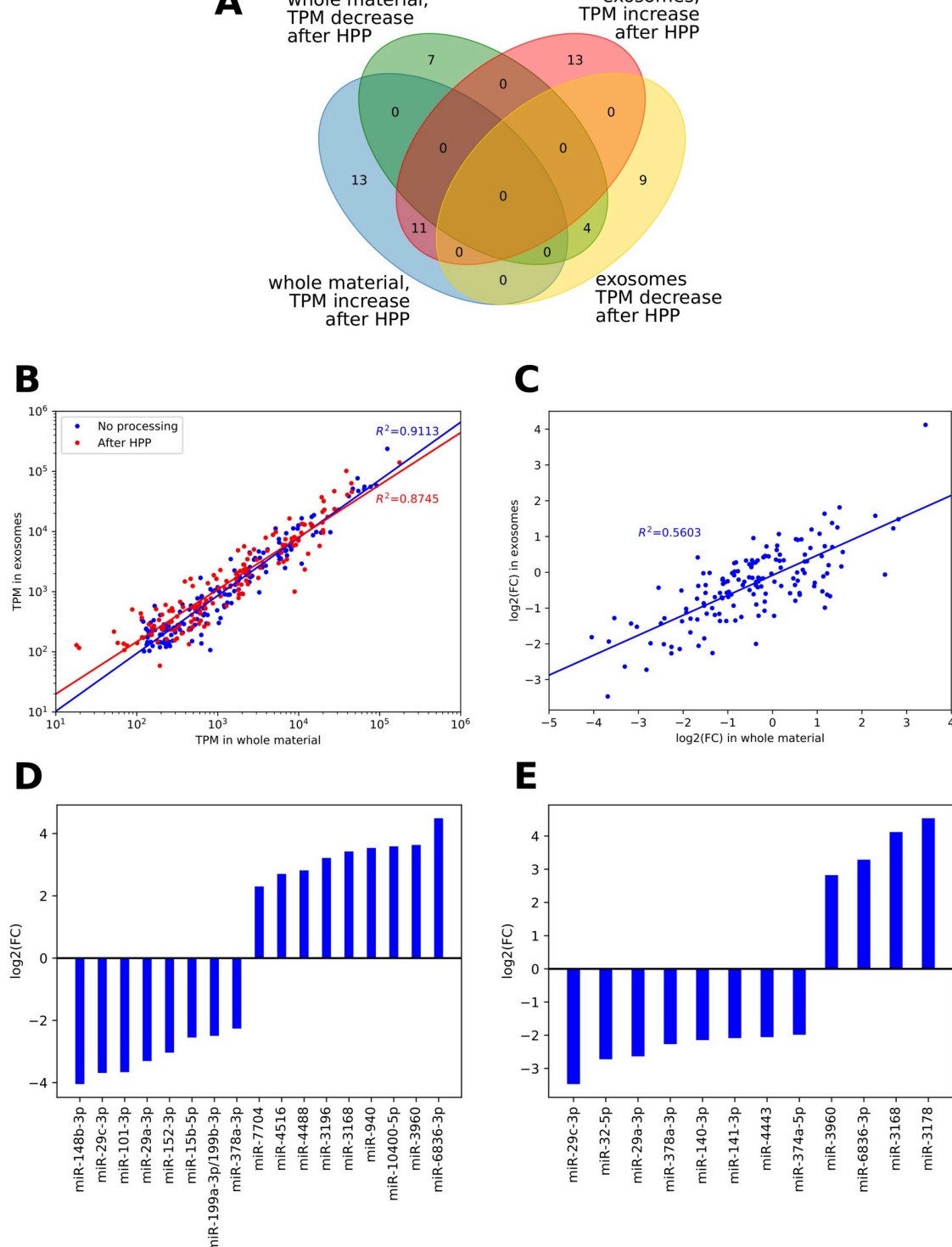

**Fig 3. Changes in miRNAs abundance resulting from HPP.** A: Venn diagram showing changes in miRNAs TPM value after HPP in whole material and exosomes; only miRNAs with significantly (FDR<0.1) differing TPM values, according to DeSEQ2 were presented; B: correlation between miRNA abundance in whole material and exosomes before and after HPP; included only miRNA with TPM at least 100 in both unprocessed materials; C: correlation between fold change in total and exosomal fraction, filtration as in panel B; D: fold change values calculated by DeSEQ2 for miRNA differentially abundant in whole material before and after HPP; included only miRNAs with TPM at least 100 before HPP; E: fold change calculated by DeSEQ2 for miRNA differentially abundant in exosomes before and after HPP; included only miRNAs with TPM at least 100 before HPP.

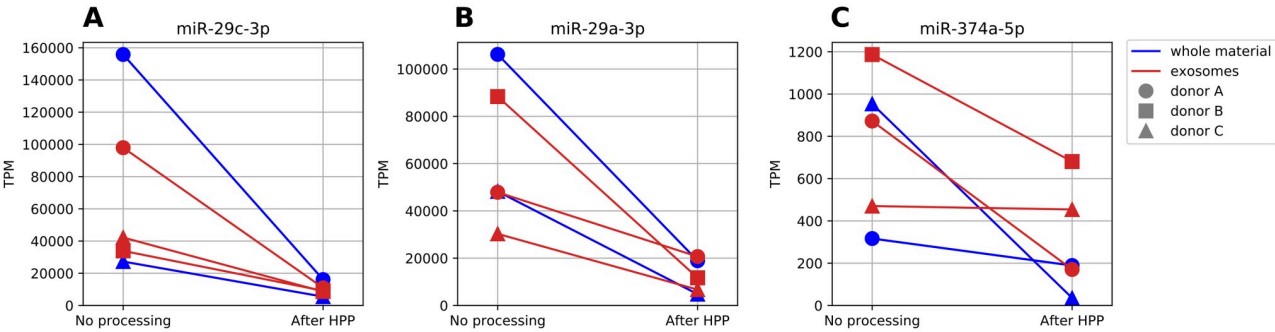

**Fig 4. Changes in abundance of miRNAs resulting from HPP, consequently observed in both whole milk and isolated exosomes.** A: miR-29c-3p; B: miR-29a-3p; C: miR-374a-5p.

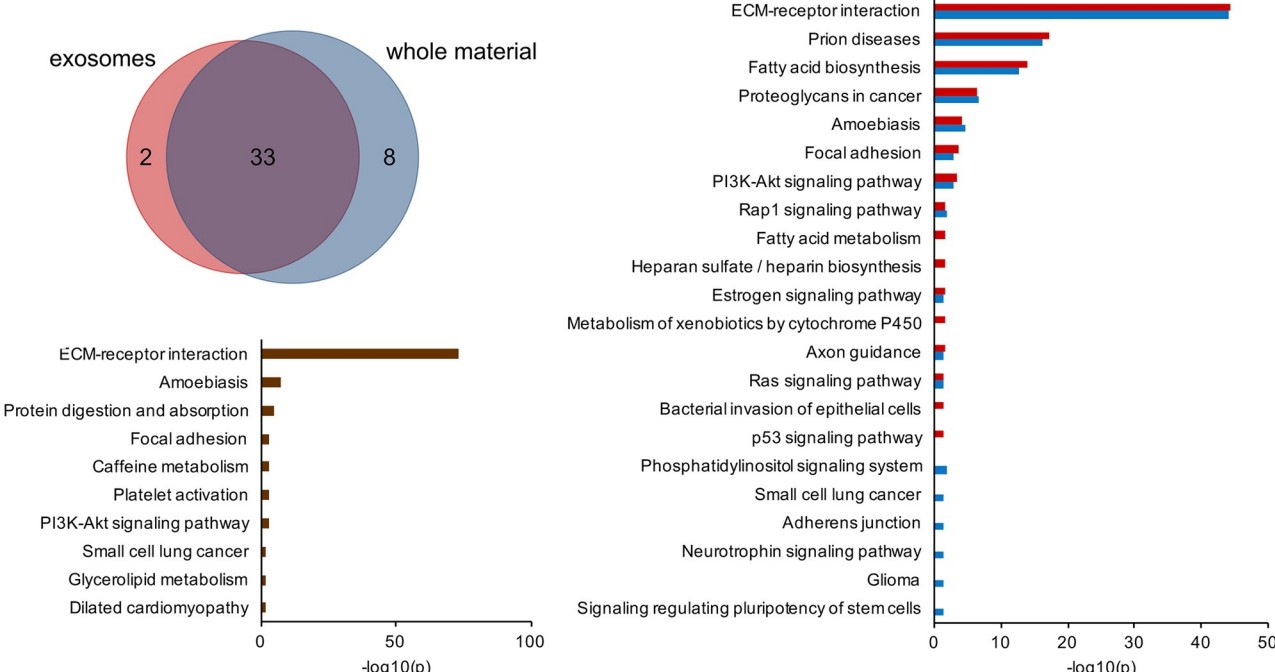

**Fig 5. Functional analysis of human milk miRNA.** A: Venn diagram showing overlap between the most abundant miRNAs accounting for 90% of whole material and exosomal reads in unprocessed milk; B: functional annotations of targets of miRNAs which were the most abundant in unprocessed milk (whole material in blue, exosomes in red); C: functional annotations of targets of 8 miRNAs which suffered the biggest depletion during HPP (miR-29c-3p, miR-32-5p, miR-29a-3p, miR-378a-3p, miR-140-3p, miR-141-3p, miR-4443, miR-374a-5p).

## Discussion

Our study showed that Holder pasteurization and high pressure processing have different impact on miRNA profile of human breast milk. The initial analysis of distribution of short RNA reads in HoP samples revealed a very high peak of reads shorter than 9 bp and nearly complete loss of inserts of length typical for miRNA and piRNA. Degradation of miRNA subjected to increased temperature in HoP was so substantial, that we were not able to perform either quantitative or functional analysis. We suspect that this is an effect of thermal degradation of miRNA and piRNA what is consistent with studies, in which a substantial decrease of miRNA abundance was observed in heated milk [46, 47]. The hypothesis is supported by

findings from the study on artichoke, in which cooking led to reduction of plant's total RNA of about 39% [48]. Similarly, pan-frying of bovine heart and adrenals led to 20–50% reduction in the number of miRNAs detected at 10 or greater reads, however it was not a case in cooked sirloin [49]. Notably, significant degradation was observed only in 5 out of 20 analyzed arti-chokes-derived miRNAs, which might suggest uneven effect of thermal processing on individual miRNAs and might explain the reason for difference in miRNAs degradation between tissues.

HPP appeared to inflict less damage to short RNA molecules, especially piRNA-size ones which remained largely intact which corresponds with the studies demonstrating similar effects on the amount of immunoglobulins [50] and other bioactive compounds [22]. Micro-RNA reads, although highly diminished, were still present in detectable amounts after HPP and according to our results exosomal sequestration seems to protect miRNA against elevated pressure, while it does not prevent thermal degradation. The ability to limit the unfavorable impact of HPP on miRNA appears as another interesting capability of milk exosomes apart from previously described in vitro resistance to digestive fluids [51] and stability in household milk storage conditions [52].

Analysis of a list of miRNAs detected in whole unprocessed milk material and in exosomes showed a significant overlap between the two. Top 10 most abundant miRNAs account for about 65–70% of the exosomal and whole milk miRNA content. These numbers matched results of a previous study showing 10 miRNA, which comprised 62% of exosomal miRNA [46]. The most abundant milk miRNA in our study—miR-148a-3p—was recently shown to regulate cell proliferation when delivered to cell culture in the form of milk-derived exosomes [53]. It constitutes almost 25% of miRNAs content in exosomal fraction and 12.4% in whole milk, and so may exert considerable impact. Other abundantly expressed miRNAs included: miR-30d-5p, let-7a-5p, let-7b-5p, which together with miR-148a-3p have been reported to be the most expressed exosomal miRNAs not only in human milk, but also in breast milk of other mammals [54–56]. Moreover, miR-148a, miR-30d and miR-200c, also present in our top 10, have been proposed to serve as biomarkers of milk quality [57].

Loss of miRNA due to HPP seems to be heterogeneous, similarly to a report by Zhou et al. [46]. miR-30d-5p appeared to be highly resistant to unfavorable conditions and ended up as the most abundant in both whole material and exosomes after HPP. In contrast, miR-148a-3p, miR-29c-3p, miR-29a-3p and miR-378-3p were affected by pasteurization in a much higher degree. We also noted miRNAs that were largely unaltered in terms of relative abundance (miR-200c-3p, let-7b-5p, miR-200a-3p). The differences in miRNA stability under harsh conditions observed in our study are parallel with ones reported by Howard et al., who measured miR-29b and miR-200c levels in bovine milk before and after heating in a microwave oven [34]. The concentration of miR-200c did not decrease, while miR-29b (representant of the same family as miR-29a and miR-29c in our study) [58] was lowered by 40% [34].

Functional pathways analysis of targets of most abundant milk miRNA identified several pathways associated with immunomodulation, which is a function of the human milk miRNA that has been widely discussed in the literature [4, 32, 59]. Moreover, pathways associated with metabolism, cells adhesion, and signaling critical for fundamental cellular functions such as proliferation and cell cycle controlling, as well as tissue development were also linked to the miRNAs detected in unprocessed milk. The most significant was the ECM-receptor interactions pathway, whose constituents are involved in organ and tissue morphogenesis, as well as maintenance of tissue structure and function [60]. Apart from the obvious impact on a growing infant, these functions may be also vital for the mammary gland itself and can reflect its current functional needs [61], since miRNA in human milk originate mainly from mammary epithelium [16].

Functional annotations of targets of miRNAs most depleted during HPP revealed which of previously identified functions were most affected by HPP. The most significant ones were pathway involved in ECM-receptor interactions, cell proliferation and metabolism regulation. These functions are similar to the roles of the majority of raw human milk's miRNA, so we conclude that HPP weakens functions of human milk by unevenly decreasing numbers of miRNAs. The depletion of miRNA involved in all those pathway is not complete, so the range of their functions is probably not altered. It is parallel with observations from other studies, showing that HPP leaves many bioactive molecules, including cytokines, immunoglobulins and lactoferrin largely preserved [21, 28, 30, 31, 62].

The main limitation of our study is the sample size. Milk samples were obtained from only 3 donor at a single timepoint, since the primary purpose of the experiment was the analysis of effect of processing on milk composition and each sample was examined 6 times (miRNA from whole milk and exosomes in unprocessed, HPP and pasteurized samples). Composition of breast milk presents in general limited inter-subject variability [63] and analysis was planned in paired samples scenario, thus limiting sample size was considered justified. Our experimental design assumed analysis of mature milk whose nutritional values change over time to much lesser extend than in the case of colostrum or milk in first few weeks post partum (translational milk) [3, 64].

## Conclusion

To conclude, we showed that HPP is less detrimental to miRNAs in human milk than HoP. Although miRNAs degrade under high pressure unevenly, the spectrum of their biological functions remains largely intact. Considering accumulating evidence of functional role of dietary miRNA, it seems reasonable to preferentially use breast milk processing techniques that preserve them.

## Supporting information

**S1 Fig. Statistics on reads not mapped to miRBase.**
(TIF)

**S1 Table. Samples data.**
(DOCX)

**S2 Table. Raw miRNA counts in all samples.**
(XLSX)

**S3 Table. Results of differential expression analysis performed in DESeq2.**
(XLSX)

**S4 Table. The most abundant miRNA accounting for 90% of whole material and exosomal reads in unprocessed milk.**
(DOCX)

## Acknowledgments

We would like to thank all the human milk donors from the Regional Human Milk Bank who participated in the study for their support and donation of biological material.

## Author Contributions

**Conceptualization:** Aleksandra Wesolowska, Wojciech Fendler.

**Data curation:** Wojciech Fendler.

**Formal analysis:** Urszula Smyczynska, Marcin M. Stanczak.

**Funding acquisition:** Marcin A. Bartlomiejczyk, Pawel Sztromwasser, Aleksandra Wesolowska, Wojciech Fendler.

**Investigation:** Marcin A. Bartlomiejczyk, Aleksandra Wesolowska, Olga Barbarska, Emilia Pawlikowska.

**Methodology:** Urszula Smyczynska, Marcin A. Bartlomiejczyk.

**Project administration:** Wojciech Fendler.

**Resources:** Marcin A. Bartlomiejczyk, Pawel Sztromwasser, Aleksandra Wesolowska, Olga Barbarska, Emilia Pawlikowska, Wojciech Fendler.

**Software:** Pawel Sztromwasser.

**Supervision:** Wojciech Fendler.

**Validation:** Marcin M. Stanczak.

**Visualization:** Urszula Smyczynska, Marcin M. Stanczak.

**Writing – original draft:** Urszula Smyczynska, Marcin A. Bartlomiejczyk, Marcin M. Stanczak.

**Writing – review & editing:** Urszula Smyczynska, Marcin A. Bartlomiejczyk, Marcin M. Stanczak, Pawel Sztromwasser, Aleksandra Wesolowska, Olga Barbarska, Emilia Pawlikowska, Wojciech Fendler.

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
