## [Decision Letter · Decision Letter 0]

19 May 2020

PONE-D-20-07569

Impact of processing method on donated human breast milk microRNA content

PLOS ONE

Dear Dr. Smyczynska,

Thank you for submitting your manuscript to PLOS ONE. After careful consideration, we feel that it has merit but does not fully meet PLOS ONE’s publication criteria as it currently stands. Therefore, we invite you to submit a revised version of the manuscript that addresses the points raised during the review process.

We would appreciate receiving your revised manuscript by Jul 03 2020 11:59PM. To enhance the reproducibility of your results, we recommend that if applicable you deposit your laboratory protocols in protocols.io, where a protocol can be assigned its own identifier (DOI) such that it can be cited independently in the future. For instructions see: http://journals.plos.org/plosone/s/submission-guidelines#loc-laboratory-protocols

We look forward to receiving your revised manuscript.

Kind regards,

Tao Huang

Academic Editor

PLOS ONE

Journal Requirements:

2. In your Methods section, please provide additional information about the participant recruitment method and the demographic details of your participants. Please ensure you have provided sufficient details to replicate the analyses such as: a) the recruitment date range (month and year), b) a description of any inclusion/exclusion criteria that were applied to participant recruitment, c) a table of relevant demographic details, d) a statement as to whether your sample can be considered representative of a larger population, e) a description of how participants were recruited, and f) descriptions of where participants were recruited and where the research took place.

Reviewers' comments:

Reviewer's Responses to Questions

**Comments to the Author**

1. Is the manuscript technically sound, and do the data support the conclusions?

Reviewer #1: Partly

Reviewer #2: Yes

Reviewer #3: Yes

2. Has the statistical analysis been performed appropriately and rigorously? 

Reviewer #1: Yes

Reviewer #2: N/A

Reviewer #3: Yes

3. Have the authors made all data underlying the findings in their manuscript fully available?

Reviewer #1: Yes

Reviewer #2: Yes

Reviewer #3: No

4. Is the manuscript presented in an intelligible fashion and written in standard English?

Reviewer #1: Yes

Reviewer #2: No

Reviewer #3: Yes

5. Review Comments to the Author

Reviewer #1: First I want to appreciate the investigators for the great work done. This study is an incredible assessment for miRNAs and the functioning in infants. This study highlights great advances in breast milk donation practices and much required knowledge in this field.

My review is based on the questions above;

1. The manuscript is generally sound but there is a great need for increment in sample size, the analysis proved that need but ultimately the knowledge has been well explained.

2. The interplay of analysis between function and composition based on the processing method was well elaborated. This was one of my strongest points.

3. The data was made available. More analysis on function in relation to miRNAs especially on the processes can be generated if possible.

4. Standard English was used. I believe they however meant "revealed" and not revelled in the discussion concerning HPP on line 328. This can be clarified please.

I would love to encourage the authors to explore more information on the composition of breast milk in relation to other products after the preservation method used.

Thank you

Reviewer #2: This manuscript by Smyczynska et al describes the differences of two different processing methods on donated human breast milk on microRNA level. This subject is of high interest and the analysis of influence of HoP and HPP on micro RNAs is unique although many publications already claimed that HPP preserves proteins and other nutrients better than HoP.

The paper is well structured and has a good readability although I would recommend a native English reviewer. The main weakness in this study I see in the sample size. Only three samples were analyzed and one sample (B) was additionally excluded. At least I want to see a critical discussion about this fact. Nevertheless, they supported the theory for using HPP processing in a well structured and clear way and supported it with nice and clear figures.

Other comments:

L28: et al. instead of at al.

L43: Recommendations From, no capital letters

L117: six instead of 6

L167 and L328: revealed instead of reveled

Reviewer #3: The authors of the manuscript compared the total and exosome-bound content of small noncoding RNAs (miRNAs) in human milk depending on two preservation methods (HoP and HPP). Authors showed that HPP is less detrimental to human milk miRNAs than HoP and thus has a potential as a processing procedures for human milk banks.

In general, the comparative analysis is very essential because could have an impact on the miRNA level in infants.

However, I have only small concerns about the study design, and believe that some additional experiments or data are required to support the conclusions.

Unfortunately, I was not able to find the access the sequences of the study. Please submit the .fastq data from miRNA to https://www.ebi.ac.uk/ena.

Authors used milk samples, obtained on the 50th day of lactation. Why not the 90th day or colostrum samples? Authors should explain the selection.

Do you have any information about the bacterial composition of the investigated milk (unprocessed vs HPP samples)? If yes, please provide or discuss. Latest studies as reviewed by Simpson et al, 2015 https://www.ncbi.nlm.nih.gov/pmc/articles/PMC4682386/ have highlighted that the expression of miRNAs is profoundly impacted by a variety of bacterial pathogens and that likewise miRNAs impose strong pressure to the invading microorganisms.

6. PLOS authors have the option to publish the peer review history of their article (what does this mean?). If published, this will include your full peer review and any attached files.

Reviewer #1: Yes: Dr Okurut Emmanuel

Reviewer #2: No

Reviewer #3: No

---

## [Author Response · Author response to Decision Letter 0]

23 Jun 2020

Dear Editors and Reviewers,

Thank you for consideration of our manuscript and your time and effort put into making it more accessible and informative. We introduced the suggested corrections wherever it was possible and discuss our rationale for taking specific course of action. Please find responses to particular comments in the list below. As several points of the reviews are repeated we have answered them in jointly, marking with respective numbers the reviews which they address

Editorial comments:

In your Methods section, please provide additional information about the participant recruitment method and the demographic details of your participants. Please ensure you have provided sufficient details to replicate the analyses such as: a) the recruitment date range (month and year), b) a description of any inclusion/exclusion criteria that were applied to participant recruitment, c) a table of relevant demographic details, d) a statement as to whether your sample can be considered representative of a larger population, e) a description of how participants were recruited, and f) descriptions of where participants were recruited and where the research took place.

Reviewer #1: First I want to appreciate the investigators for the great work done. This study is an incredible assessment for miRNAs and the functioning in infants. This study highlights great advances in breast milk donation practices and much required knowledge in this field.

My review is based on the questions above;

1. The manuscript is generally sound but there is a great need for increment in sample size, the analysis proved that need but ultimately the knowledge has been well explained.

2. The interplay of analysis between function and composition based on the processing method was well elaborated. This was one of my strongest points.

3. The data was made available. More analysis on function in relation to miRNAs especially on the processes 

can be generated if possible.

4. Standard English was used. I believe they however meant "revealed" and not revelled in the discussion concerning HPP on line 328. This can be clarified please.

Reviewer #2: This manuscript by Smyczynska et al describes the differences of two different processing methods on donated human breast milk on microRNA level. This subject is of high interest and the analysis of influence of HoP and HPP on micro RNAs is unique although many publications already claimed that HPP preserves proteins and other nutrients better than HoP.

The paper is well structured and has a good readability although I would recommend a native English reviewer. The main weakness in this study I see in the sample size. Only three samples were analyzed and one sample (B) was additionally excluded. At least I want to see a critical discussion about this fact. Nevertheless, they supported the theory for using HPP processing in a well structured and clear way and supported it with nice and clear figures.

Other comments:

L28: et al. instead of at al.

L43: Recommendations From, no capital letters

L117: six instead of 6

L167 and L328: revealed instead of reveled

Reviewer #3: The authors of the manuscript compared the total and exosome-bound content of small noncoding RNAs (miRNAs) in human milk depending on two preservation methods (HoP and HPP). Authors showed that HPP is less detrimental to human milk miRNAs than HoP and thus has a potential as a processing procedures for human milk banks.

In general, the comparative analysis is very essential because could have an impact on the miRNA level in infants.

However, I have only small concerns about the study design, and believe that some additional experiments or data are required to support the conclusions.

Unfortunately, I was not able to find the access the sequences of the study. Please submit the .fastq data from miRNA to https://www.ebi.ac.uk/ena.

Authors used milk samples, obtained on the 50th day of lactation. Why not the 90th day or colostrum samples? Authors should explain the selection.

Do you have any information about the bacterial composition of the investigated milk (unprocessed vs HPP samples)? If yes, please provide or discuss. Latest studies as reviewed by Simpson et al, 2015 https://www.ncbi.nlm.nih.gov/pmc/articles/PMC4682386/ have highlighted that the expression of miRNAs is profoundly impacted by a variety of bacterial pathogens and that likewise miRNAs impose strong pressure to the invading microorganisms.

Responses:

RE&R1: In response to the Editor’s and Reviwer’s requests, available data about milk donors were included in the manuscript. All 3 recruited women were on the regular basis the milk donors to the Regional Human Milk Bank in Holy Family Hospital in Warsaw, Poland. They were healthy and fulfilled the requirements of becoming milk donors (no addictions and excluded severe chronic diseases).

R1&R2: The low number of samples, pointed by Reviewer #1 and Reviewer #2, was elaborated in depth to the discussion as the limitation of the study. Our main explanation of using only 3 milk samples is the number of analysed conditions for each of them (3 processing methods and 2 types of material – whole milk and exosomes, yielding 6 assays per sample in total). Having only 3 samples, it is difficult to determine with certainty if they are representative for whole population. However, we do not expect our samples to significantly differ from typical, normal milk composition, since the volunteers’ health was sufficiently good to allow them to become milk donors.

R1&R3: Milk samples were obtained at 50th day of lactation, because we were interested in investigating mature milk that changes less over time then early milk (colostrum or transitional milk). According to the literature at the 50th day of lactation milk is already fully mature and we should not expect much different results if it had been donated later. A similar comment was added to the manuscript as per Reviewer’s #3 request.

R1: The literature on the effect of food processing on miRNA content was added to relevant sections as suggested by Reviewer #1. Such studies are however very limited in number and usually concern thermal processing methods, while the High Pressure Processing of dietary products is much less studied. Diary products were analysed most often and results vary, but the general consensus seems to be that elevated temperature leads to partial or complete degeneration of miRNA in food.

R3 Bacterial composition of milk treated with either pasteurization method was not analysed in our study, since previous studies showed that both Holder Pasteurization and HPP effectively inactivate microorganisms that may be present in food. We know that the volunteers had not presented any symptoms of infections, including mammary gland infections when milk was obtained as per requirements posed to milk donors in our milk bank. We agree with Reviewer #3 that analysis of effect of bacteria on miRNA profile of breast milk would be interesting, but it would require substantially more samples and a completely different study design most likely focusing on a large cohort of breastfeeding mothers to evaluate the population variability of bacteria colonizing the breast and their potential to spread onto expressed milk. While undoubtedly interesting, such a study would be highly divergent from what we could do with the current design.

R1&R3 The sequences from our study in FASTQ files had already been deposited in NCBI Sequence Read Archive (SRA) under the accession number SRP238092 and this data is linked to our Gene Expression Omnibus (GEO) entry identified by accession number GSE142282. We support open science and sharing research data, but we believe that it is enough to use one repository where data is freely available to research community.

R1&R2 Typesetting errors, indicated by all reviewers, were corrected. The only exception is line 43 where we left “Recommendations From” with “From” starting with capital letter, since it is used in this form in the document that is cited there.

Should you need any more information feel free to contact me at your convenience at wojciech_fendler@dfci.harvard.edu.

Wojciech Fendler

---

## [Decision Letter · Decision Letter 1]

30 Jun 2020

Impact of processing method on donated human breast milk microRNA content

PONE-D-20-07569R1

Dear Dr. Smyczynska,

We’re pleased to inform you that your manuscript has been judged scientifically suitable for publication and will be formally accepted for publication once it meets all outstanding technical requirements.

Kind regards,

Tao Huang

Academic Editor

PLOS ONE

Additional Editor Comments (optional):

Reviewers' comments:

Reviewer's Responses to Questions

**Comments to the Author**

1. If the authors have adequately addressed your comments raised in a previous round of review and you feel that this manuscript is now acceptable for publication, you may indicate that here to bypass the “Comments to the Author” section, enter your conflict of interest statement in the “Confidential to Editor” section, and submit your "Accept" recommendation.

Reviewer #2: All comments have been addressed

Reviewer #3: All comments have been addressed

2. Is the manuscript technically sound, and do the data support the conclusions?

Reviewer #2: Yes

Reviewer #3: Yes

3. Has the statistical analysis been performed appropriately and rigorously? 

Reviewer #2: Yes

Reviewer #3: Yes

4. Have the authors made all data underlying the findings in their manuscript fully available?

Reviewer #2: Yes

Reviewer #3: Yes

5. Is the manuscript presented in an intelligible fashion and written in standard English?

Reviewer #2: Yes

Reviewer #3: Yes

6. Review Comments to the Author

Reviewer #2: The authors addressed my comments and changed inaccuracies carefully. The argument on the low sample-size was highlighted and spelling mistakes were corrected. Also the comments of the other authors were addressed. Great work was done on an important topic.

Reviewer #3: (No Response)

7. PLOS authors have the option to publish the peer review history of their article (what does this mean?). If published, this will include your full peer review and any attached files.

Reviewer #2: **Yes: **Christian Robben

Reviewer #3: No

---

## [Editor Report · Acceptance letter]

6 Jul 2020

PONE-D-20-07569R1 

Impact of processing method on donated human breast milk microRNA content 

Dear Dr. Smyczynska:

I'm pleased to inform you that your manuscript has been deemed suitable for publication in PLOS ONE. Congratulations! Your manuscript is now with our production department. 

Kind regards, 

on behalf of

Dr. Tao Huang 

Academic Editor

PLOS ONE